# Stabilization and structural analysis of a membrane-associated hIAPP aggregation intermediate

**Diana C Rodriguez Camargo[1,2,3†], Kyle J Korshavn[2†], Alexander Jussupow[1], Kolio Raltchev[3], David Goricanec[3], Markus Fleisch[4], Riddhiman Sarkar[3], Kai Xue[4], Michaela Aichler[4], Gabriele Mettenleiter[4], Axel Karl Walch[4], Carlo Camilloni[1], Franz Hagn[1,3,4], Bernd Reif[3,4], Ayyalusamy Ramamoorthy[1,2]***

[1]Institute for Advanced Study, Technische Universität München, Garching, Germany; [2]Program in Biophysics, Department of Chemistry, University of Michigan, Ann Arbor, United States; [3]Center for Integrated Protein Science Munich (CIPSM), Department Chemie, Technische Universität München, Garching, Germany; [4]Helmholtz Zentrum München, Neuherberg, Germany

**Abstract** Membrane-assisted amyloid formation is implicated in human diseases, and many of the aggregating species accelerate amyloid formation and induce cell death. While structures of membrane-associated intermediates would provide tremendous insights into the pathology and aid in the design of compounds to potentially treat the diseases, it has not been feasible to overcome the challenges posed by the cell membrane. Here, we use NMR experimental constraints to solve the structure of a type-2 diabetes related human islet amyloid polypeptide intermediate stabilized in nanodiscs. ROSETTA and MD simulations resulted in a unique β-strand structure distinct from the conventional amyloid β-hairpin and revealed that the nucleating NFGAIL region remains flexible and accessible within this isolated intermediate, suggesting a mechanism by which membrane-associated aggregation may be propagated. The ability of nanodiscs to trap amyloid intermediates as demonstrated could become one of the most powerful approaches to dissect the complicated misfolding pathways of protein aggregation.
DOI: https://doi.org/10.7554/eLife.31226.001

**\*For correspondence:**
ramamoor@umich.edu

[†]These authors contributed equally to this work

**Competing interests:** The authors declare that no competing interests exist.

## Introduction

Protein aggregation and amyloid formation have been implicated in a range of human pathologies, including Alzheimer's disease (AD), Parkinson's disease, and type II diabetes (*Chiti and Dobson, 2017*; *Hartl, 2017*). While the disease phenotypes and the implicated proteins or peptides differ widely, the associated aggregation phenomenon and amyloid formation often have many commonalities such as the role of cell membrane in catalyzing the generation of toxic intermediates. Many of these proteins have been observed to interact preferentially with cellular membranes which may subsequently promote unique folded structures and/or promote amyloid formation while simultaneously altering membrane composition, structure, and integrity (*Aisenbrey et al., 2008*; *Byström et al., 2008*). Structural insights into the interaction of α-synuclein, an amyloidogenic peptide associated with Parkinson's disease, with membrane have been facilitated by the propensity for α-synuclein to readily adopt a helical conformation in the presence of lipids as well as the relatively slow rates of α-synuclein aggregation (*Fusco et al., 2014*). Other amyloidogenic peptides, such as amyloid-β (Aβ) or human islet amyloid polypeptide (hIAPP), have been less amenable to high-resolution structural analysis in the presence of membrane, possibly due to their rapid aggregation and membrane disrupting effects, lower propensity towards structure on the membrane, or increased structural heterogeneity.

Some insights have been gleaned regarding early, transient interactions between monomeric Aβ and lipid a bilayer (*Korshavn et al., 2016*), along with preliminary insights into Aβ aggregates prepared at either exceptionally high peptide concentrations (*Delgado et al., 2016*) or in the presence of detergents which can dramatically impact peptide structure (*Serra-Batiste et al., 2016*). The rat variant of hIAPP (rIAPP), which does not form amyloid fibrils and not toxic under most conditions, has been used to generate models of membrane-associated dimers (*Nath et al., 2011*). This structure was then successfully used to screen for small molecules which promote membrane-associated toxicity of hIAPP (*Nath et al., 2015*). While this structure reaffirms the usefulness of mimetic peptides in the study of amyloids in general, the study of native, amyloidogenic sequences in the presence of membrane remains extremely challenging.

In order to better study integral membrane proteins in a near-native lipid bilayer environment, recent studies have reported the successful applications of lipid nanodiscs. These nanodiscs traditionally consist of a small (~8–15 nm in diameter), circular patch of lipids surrounded by a scaffold protein, peptide, or polymer and facilitate the stable reconstitution of membrane proteins in their near-native environment (*Hagn et al., 2013*). Nanodiscs have previously been used to study the native function of full-length membrane proteins, protein-protein interactions between integral membrane proteins, and to generate structural data of the typically difficult class of proteins (*Denisov and Sligar, 2016*). Nanodiscs were also utilized in a previous study of a stabilized rIAPP dimer (*Nath et al., 2011*). Due to the constrained size of the lipid bilayer and devoid of curvature, it is likely that peptide aggregation on the flat surface will be restricted after reaching a certain aggregate size, unlike the aggregation on the surface of a lipid vesicle which is relatively unconstrained and may therefore progress to elongated fibers characteristic of amyloids (*Aisenbrey et al., 2008*; *Zhang et al., 2017*). Small, isotropic nanodiscs, optimal for solution NMR applications, have also been developed; these nanodisc variants are ideal for the structural analysis of the anticipated stabilized intermediate which may be analyzed in a similar manner as shown previously with integral membrane proteins (*Hagn et al., 2013*).

Here, we evaluated hIAPP, a 37-residue model amyloidogenic peptide, in order to explore the ability of lipid nanodiscs to stabilize distinct, membrane-associated amyloid oligomers. hIAPP aggregation is strongly associated with the progression of type II diabetes (*Westermark et al., 1987*). Furthermore, its aggregation on lipid bilayers has been previously demonstrated to destabilize the membrane through multiple mechanisms, suggesting the existence of discrete, non-fibrillar oligomeric species which may be pathogenic and potential targets for isolation via nanodisc stabilization (*Brender et al., 2012*). Similar to many other amyloids, hIAPP aggregation kinetics and intermediates depend on both the solution conditions and membrane composition; nanodisc-mediated stabilization of folded intermediates may also exhibit a similar dependency. Thus, a thioflavin-T (ThT)-based fluorescence screen was initially used to characterize hIAPP aggregation in the presence of three different membrane scaffold protein-based nanodisc compositions (*Table 1*) and buffer conditions (*Hagn et al., 2013*). Varying the ratio of negatively charged phosphatidylglycerol (PG) and zwitterionic phosphatidylcholine (PC) lipids may tune the affinity of hIAPP for the nanodisc surface (*Zhang et al., 2017*). Temperature was also modulated to alter the bilayer fluidity, which has previously been demonstrated to modulate the ability of peptides to insert into lipid bilayers (*Barrera et al., 2012*; *Sani et al., 2012*). Finally, the effect of solution pH on hIAPP aggregation in the presence of various nanodiscs was analyzed given the ability of slightly lower pH to dramatically alter hIAPP's aggregation behavior (*Jha et al., 2014*). The optimal combination of

**Table 1.** Nanodisc identity and composition.
All nanodiscs were formed at a protein (MSP):lipid ratio of 1:50 and purified by size exclusion chromatography prior to use.

| Nanodisc | Lipid composition |
| --- | --- |
| ND1 | 90% DMPC/10% DMPG |
| ND2 | 75% DMPC/25% DMPG |
| ND3 | 50% DMPC/50% DMPG |

DOI: https://doi.org/10.7554/eLife.31226.002

nanodisc composition, temperature, and solution pH was subsequently subjected to biochemical characterization and structural analysis by NMR. Through chemical shift analysis, we identified, for the first time, a non-fibrillar β-sheet conformation of hIAPP directly associated with the nanodisc lipid bilayer. This represents the first high-resolution structural model based on experimental constraints of hIAPP associated with a native lipid bilayer and demonstrates the great potential of nanodiscs as a tool to trap and stabilize membrane-associated aggregates of amyloidogenic peptides and proteins in a native, planar bilayer environment.

## Results

### Assembly kinetics of hIAPP with nanodiscs

ThT is a ubiquitous fluorescent probe in the interrogation of amyloid aggregation kinetics and mechanisms, and it is commonly used to characterize the aggregation of various amyloidogenic peptides in the presence of lipid bilayers, making ThT an ideal tool for the initial identification of a lipid bilayer and buffer system suitable for the stabilization and subsequent structural characterization of a membrane-associated hIAPP intermediate (*Galvagnion et al., 2015*; *Zhang et al., 2017*). After verifying that the fluorescent properties of ThT are minimally perturbed by the presence of nanodiscs in solution (*Figure 1a*), a suite of conditions, including varied lipid nanodisc compositions, pH, and temperature were evaluated for their ability to influence the kinetics of hIAPP aggregation as observed by ThT (*Figure 1b* and *Figure 2*). Resulting curves were subsequently fit to a logarithm to extrapolate their lag time ($t_{lag}$) which correlates to the time required for peptide to convert from its monomeric state to an aggregation-competent oligomer (*Figure 1c*) (*Batzli and Love, 2015*). If a set of conditions is capable of promoting a stable membrane-associated intermediate it is likely that the observed $t_{lag}$ will increase and/or fibrillation will be completely halted due to the newly stabilized species inhibiting aggregation.

These preliminary results revealed a number of factors regarding hIAPP-nanodisc interactions and their role in peptide aggregation. While it is known that anionic lipids accelerates fibrillation in a dose dependent manner, we observed that, under most conditions, increasing the concentration of nanodiscs increased the $t_{lag}$ and delayed aggregation (*Cao et al., 2013*). However, in agreement with previous observations, as the percentage of DMPG in the nanodisc was increased, the delay in aggregation was reduced (*Zhang et al., 2017*). When the nanodisc reached 50% DMPG, the aggregation kinetics in the presence of nanodisc were extremely similar to those in the absence, regardless of total lipid concentration. This suggests that the inhibitory ability of nanodiscs is highly dependent upon the concentration of negatively charged lipids; too high a concentration of PG abrogated any inhibitory capacity. Modulating the ratio of membrane components is capable of tuning these two components. It was also observed that raising the solution temperature from 25°C to 35°C generally enhanced the inhibitory activity of lipid nanodiscs, although increases in temperature have previously been shown to accelerate amyloid formation in solution (*Batzli and Love, 2015*). The phase transition temperature for the dimyristoyl lipids used in this study is approximately 24°C, thus elevating the temperature to 35°C ensures that the bilayer is completely fluid and may promote peptide insertion into the nanodisc, as hypothesized. Additionally, similar to aggregation experiments performed in the absence of lipid bilayers, decreasing the pH from 7.4 to 5.3 delayed hIAPP aggregation and increased the potency of nanodisc-mediated inhibition (*Jha et al., 2014*).

Based upon the ThT screening results, it was determined that utilizing ND1 (90% DMPC/10% DMPG) in acetate buffer (pH 5.3) would most likely yield a stable, nanodisc associated hIAPP intermediate. Under these conditions, regardless of the temperature studied, fibrillation was not observed, even after 2000 min. To confirm the ability of ND1 to block large aggregate formation, transmission electron microscopy (TEM) was employed (*Figure 1d*). While hIAPP incubated at pH 5.3 for 1 week generated conventional amyloid fibrils, hIAPP co-incubated with 1 equiv. of ND1 did not form large fibrillar aggregates during the same incubation time. Instead, nanodiscs of increased size, relative to peptide-free ND1, were observed. This increase in size suggests that hIAPP successfully interacted with and incorporated into ND1 to generate a larger, stable complex, similar to size increases observed for other protein complexes contained within nanodiscs (*Xu et al., 2013*). The ability of ND1 to stabilize a non-fibrillar intermediate was further investigated through solution NMR. During amyloid formation, the intensity of the observable resonances originating from the

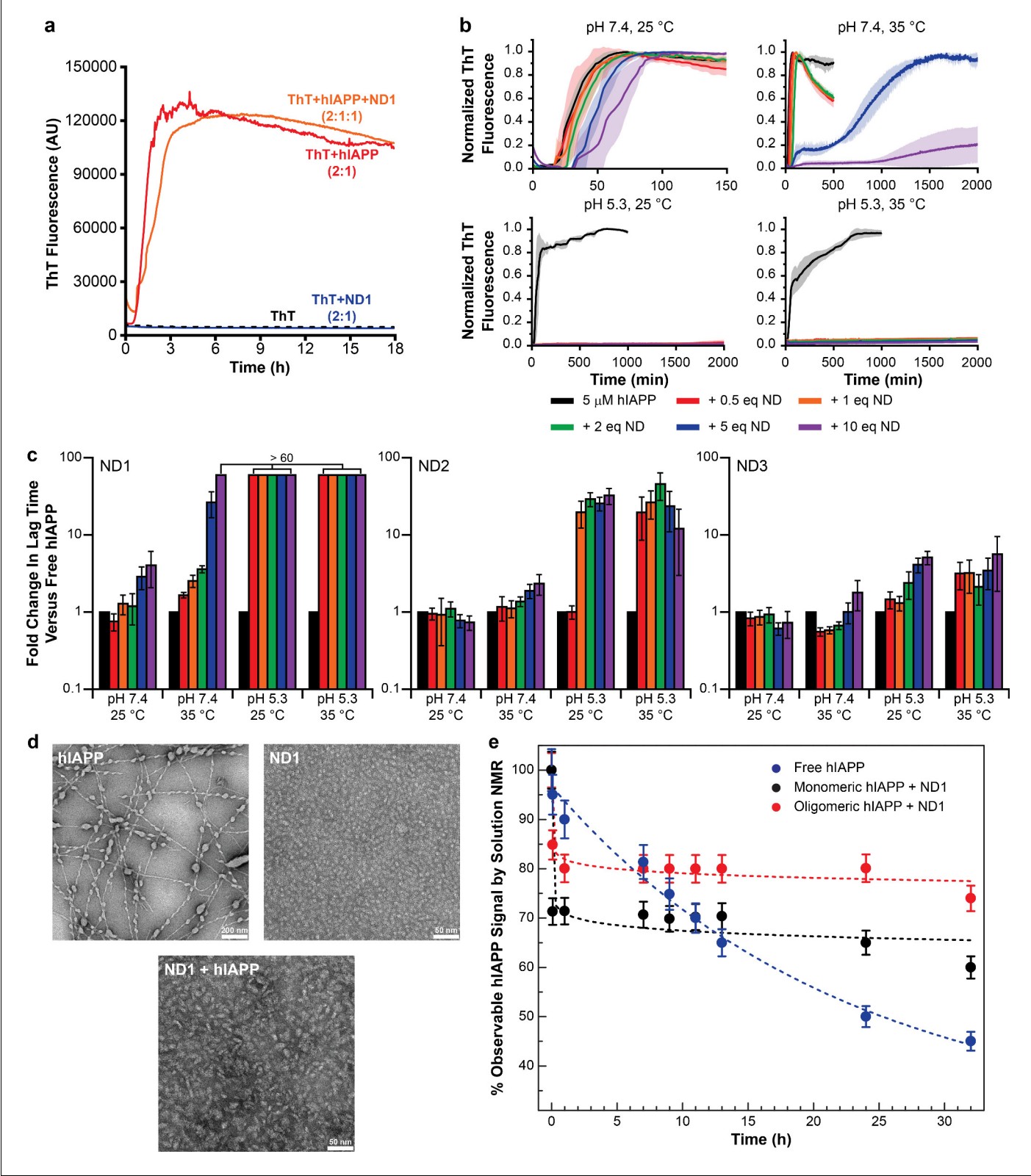

**Figure 1.** Nanodiscs modulate the kinetics of hIAPP aggregation. (a) Thioflavin T (ThT) was determined to have no interactions with nanodiscs which could significantly alter the dye's fluorescent properties (20 mM $PO_4$ pH 7.4, 50 mM NaCl, 25°C). (b) ThT fluorescence was monitored as hIAPP (5 μM) was incubated with increasing concentration of ND1 under different conditions (either 20 mM $PO_4$ pH 7.4 or 30 mM acetate pH 5.3, both with 50 mM NaCl at either 25 or 35°C). Solid curves represent the average of three independent trials while the shaded regions represent the standard deviations of

*Figure 1 continued on next page*

*Figure 1 continued*

those measurements. (c) Lag times were calculated for each individual kinetic trace for hIAPP incubated with ND1, ND2, and ND3 (*Figure 2*). The fold change in the lag time compared to untreated hIAPP are plotted with respect to both nanodisc concentration and sample conditions. (d) TEM was used to image samples of hIAPP (50 µM) fiber prepared in the absence of nanodisc, freshly prepared ND1 (50 µM), and hIAPP monomer (50 µM) incubated with ND1 (50 µM). All samples were prepared in 30 mM acetate pH 5.3 at 35°C. (e) The overall signal intensities measured from 2D $^1$H-$^{15}$N HMQC spectra of hIAPP backbone amides in the absence or presence of ND1 were monitored over time. Peptide was prepared via both a monomeric and oligomeric methods (see Materials and methods for details) prior to treating with ND1.

DOI: https://doi.org/10.7554/eLife.31226.003

monomeric protein decreased due to the formation of larger, NMR invisible aggregates (*Figure 1e*). Inhibition of this aggregation would maintain the signal from monomeric protein for an extended period of time. The NMR signal for monomeric hIAPP in solution decayed rapidly and reached 50% of its initial intensity after 25 hr. However, freshly prepared, monomeric hIAPP in the presence 1 equiv. of ND1 maintained a relative intensity of ~70% following a rapid initial decay, possibly due to early oligomer formation. These data suggest that ND1 under slightly acidic conditions is capable of blocking conventional amyloid formation by hIAPP and may successfully stabilize a membrane-associated intermediate. This combination of buffer and lipid conditions was used exclusively in subsequent analysis of hIAPP-membrane interactions.

## Stabilization and structural characterization of an hIAPP intermediate

While ND1 is capable of blocking hIAPP fibrillation, applying the optimized conditions to stabilize a distinct, highly populated intermediate state requires further optimization. The aggregation pathway and intermediates formed by amyloidogenic peptides have been previously shown to depend on conditions, particularly peptide concentration and preparation prior to experimentation (*Brender et al., 2015*; *Serra-Batiste et al., 2016*). To explore the effects of peptide preparation on unique intermediate stabilization, both freshly prepared monomer and a mixed population of oligomers were both prepared, analyzed by both size exclusion chromatography (SEC) and gel electrophoresis, and analyzed for their unique interactions with ND1 (*Figure 3a–c*). While the oligomeric preparation generated a variety of differently sized species in solution, when separated by both SEC and gel electrophoresis, a single population of hIAPP was observed when the oligomeric population was incubated with ND1, suggesting the stabilization of a unique intermediate. Additionally, dynamic light scattering (DLS) indicates that incubation of the oligomeric hIAPP with ND1 generates species with a larger hydrodynamic radius than free ND1, suggesting that hIAPP is able to interact directly with ND1 under these conditions and generate a larger, stable complex (*Figure 3d*). Additionally, when oligomeric hIAPP was incubated with ND1 and its NMR signal monitored over time, it stabilized at approximately 80% relative intensity after a rapid drop off, a trend similar to monomeric hIAPP but with a larger percentage of the signal remaining visible (*Figure 1e*). This suggests that while both preparations are capable of binding to ND1 and stabilizing non-fibrillar intermediates, they may be stabilized at different points or the size of the stabilized population may differ.

Therefore, the ability of both the monomeric and oligomeric preparations of hIAPP to interact with ND1 were further investigated by NMR experiments to better determine the differences in their structures and aggregation intermediates (*Figure 3e and f*). The $^1$H-$^{15}$N HMQC spectra for both monomeric and the oligomeric preparations appear nearly identical in the absence of ND1. Both spectra exhibit minimal dispersion and chemical shifts similar to those previously reported for monomeric hIAPP in solution (*Brender et al., 2015*; *Rodriguez Camargo et al., 2017*). It was shown by SEC that the oligomeric preparation contains a mixture of monomeric and oligomeric species; it is possible that the spectral similarity is due to the monomeric population of the oligomer preparation (*Figure 3a*). When monomeric hIAPP was incubated with 1 equiv. of ND1, minimal spectral change was seen (*Figure 3e*, black spectrum). The observable residues showed only modest chemical shift perturbations and few resonances exhibited broadening. This suggests that only a small portion of the NMR visible hIAPP population in the monomeric preparation stably interacts with ND1 within the duration of the NMR experiment (~1 hr); monomeric hIAPP undoubtedly binds to ND1; however, the exchange rate of the highly dynamic process is too rapid to result in detectable spectral changes. Additionally, a 2D $^1$H-$^{15}$N projection of a 3D triple-resonance HNCA experiment of monomeric

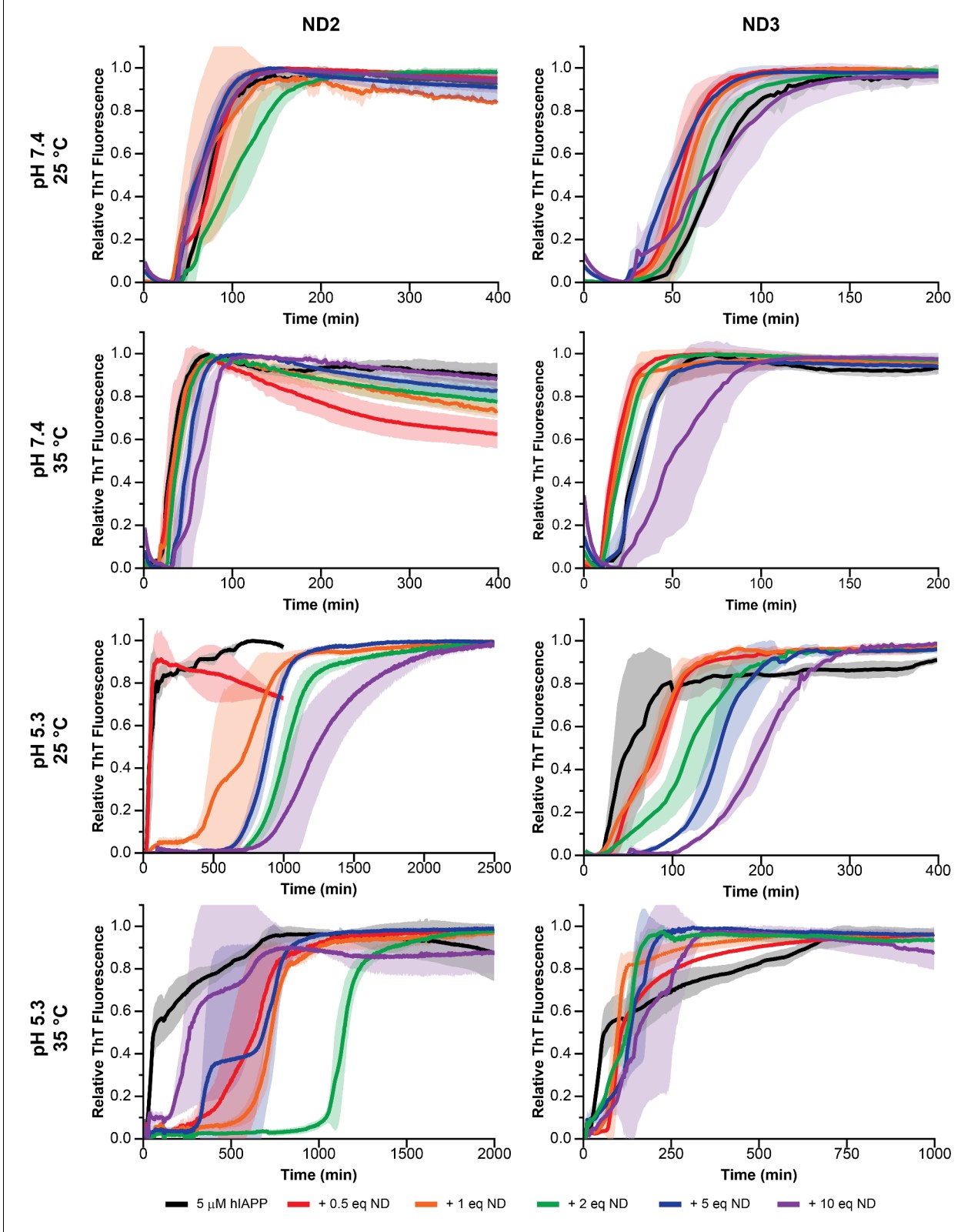

**Figure 2.** Nanodisc composition and environment dictates the extent of modulation on hIAPP aggregation. By changing the concentration of negatively charged DMPG lipids in the nanodisc (25% in ND2% and 50% in ND3), the pH of the surrounding buffer (7.4 with 20 mM PO$_4$ or 5.3 with 30 mM acetate) and the solution temperature, a wide range of kinetic behaviors can be observed for hIAPP.

DOI: https://doi.org/10.7554/eLife.31226.004

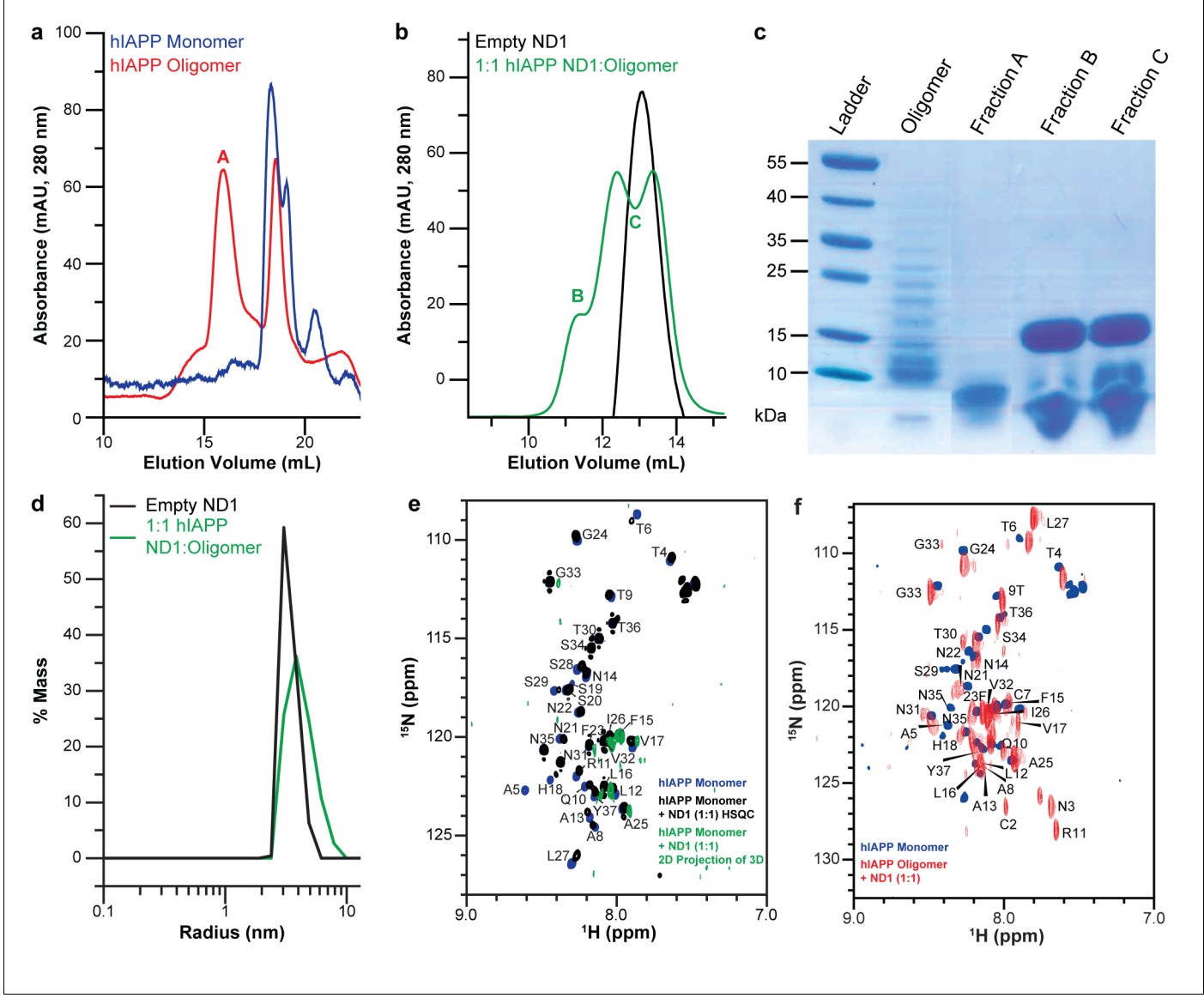

**Figure 3.** Peptide preparation impacts the stabilization of a folded hIAPP species by ND1. (**a**) Freshly dissolved hIAPP (blue) and the oligomer preparation of hIAPP (red) indicate two distinct populations of the peptide. (**b**) When the oligomeric hIAPP was incubated with ND1, a larger peptide-ND1 complex was stabilized. (**c**) Gel electrophoresis highlights changes in the oligomer population before and after incubation with ND1 and purification by SEC. (**d**) DLS confirms the findings of SEC; treatment of ND1 with oligomeric hIAPP promotes a larger, stabilized, species. (**e**) When ND1 is added to monomeric hIAPP (black), there is minimal spectral perturbation in the 2D $^{15}$N/$^1$H HMQC spectrum, suggesting minimal change in the structure. Additionally, when HNCA triple-resonance NMR experiments were performed on the same sample and the spectrum was compressed into the N-H dimensions, a dramatic reduction in signal intensity and disappearance of peaks were observed (green), further suggesting a lack of structural changes in the peptide. (**f**) Compression of the HNCA spectrum into N-H dimensions yields a full 2D spectrum with an increased dispersion, indicative of a more folded state, which can be completely assigned, facilitating further structural analysis.

DOI: https://doi.org/10.7554/eLife.31226.005

hIAPP mixed with ND1 at a 1:1 ratio showed only a few peaks, suggesting that the sample is either unstable or hIAPP exists in many distinct populations, resulting in a broadening of resonances (*Figure 3e*, green spectrum). In contrast, when the oligomeric preparation of hIAPP was incubated with ND1 at a 1:1 ratio and subjected to the same HNCA experiment, the 2D $^1$H-$^{15}$N projection spectrum showed increased signal dispersion relative to the $^1$H-$^{15}$N HMQC spectrum taken in the absence of ND1. Furthermore, a large set of resonances displayed significant chemical shift perturbations, suggesting that the NMR visible population was capable of interacting with, and potentially

inserting into ND1 in a stable manner (*Figure 3f*). Additionally, it suggested that this sample was suitable for resonance assignment by 3D NMR experiments to obtain structural insights into the new, membrane-associated intermediate using chemical shift information. In order to check the long-term sample stability, a control sample of oligomeric hIAPP with ND1 (1:10 equiv.) was prepared and monitored over the course of 1 month (*Figure 4*). While some spectral changes were evident at the end of the time course, the spectra were consistent for the majority of the experiment.

Using this optimized sample of an oligomeric preparation of hIAPP in the presence of ND1 (50 µM each) both HNCA and HNCOCA triple-resonance experiments were performed using non-uniform sampling (NUS) in order to sequentially assign the backbone resonances while utilizing a low peptide concentration (*Figure 5a* and *Figure 6*). From these assignments, backbone resonances for 30 of 37 residues were unambiguously assigned. Missing residues are predominantly located in the N- and C-termini of the hIAPP sequence. Following the assignment of 3D spectra, chemical shifts for all backbone resonances were extracted and used to calculate both the secondary structure propensity (SSP) and the $\Delta\delta^{13}C_\alpha$-$C_\beta$ secondary chemical shifts to generate secondary structure predictions for the membrane-associated folded intermediate (*Figure 5b*) (*Marsh et al., 2006*). Both SSP and $\Delta\delta^{13}C_\alpha$-$C_\beta$ predict the presence of three β-strands within a monomeric subunit of the folded species bound to ND1. This stands in stark contrast to the previously reported, partially α-helical structure predicted for hIAPP under similar conditions in the absence of lipid (*Rodriguez Camargo et al., 2017*), supporting the hypothesis that lipid nanodiscs can be applied to trap folded amyloidogenic intermediates. The structure is also markedly different from the previously reported rIAPP dimer bound to the surface of a nanodisc, reaffirming the importance of sequence and sample conditions on amyloid folding (*Nath et al., 2011*).

To further refine the model of folded hIAPP interacting with ND1, Chemical Shift-ROSETTA was used to compile all chemical shift data obtained from the 2D and 3D NMR spectra to generate an all-atom structural model by combining the 10 lowest energy structures (*Figure 5c* and *Figure 7*) (*Shen et al., 2008*, Shen et al., 2009Shen et al., 2009*Shen et al., 2009*). For the compiled structures, the $C_\alpha$-RMSD for residues 6–34, which were unambiguously assigned in 3D spectra, was 1.946 ± 0.521, while for all other residues the $C_\alpha$-RMSD was 3.534 ± 0.489 due to a lack of experimental restraints. Overall, the simulated structural model represents a consistently folded hIAPP monomeric subunit. Three antiparallel β-strands are observed for A8-L12, F15-H18, and I26-S29 with flexible loops connecting them. Multiple residues (G24, A25) associated with the amyloid-driving

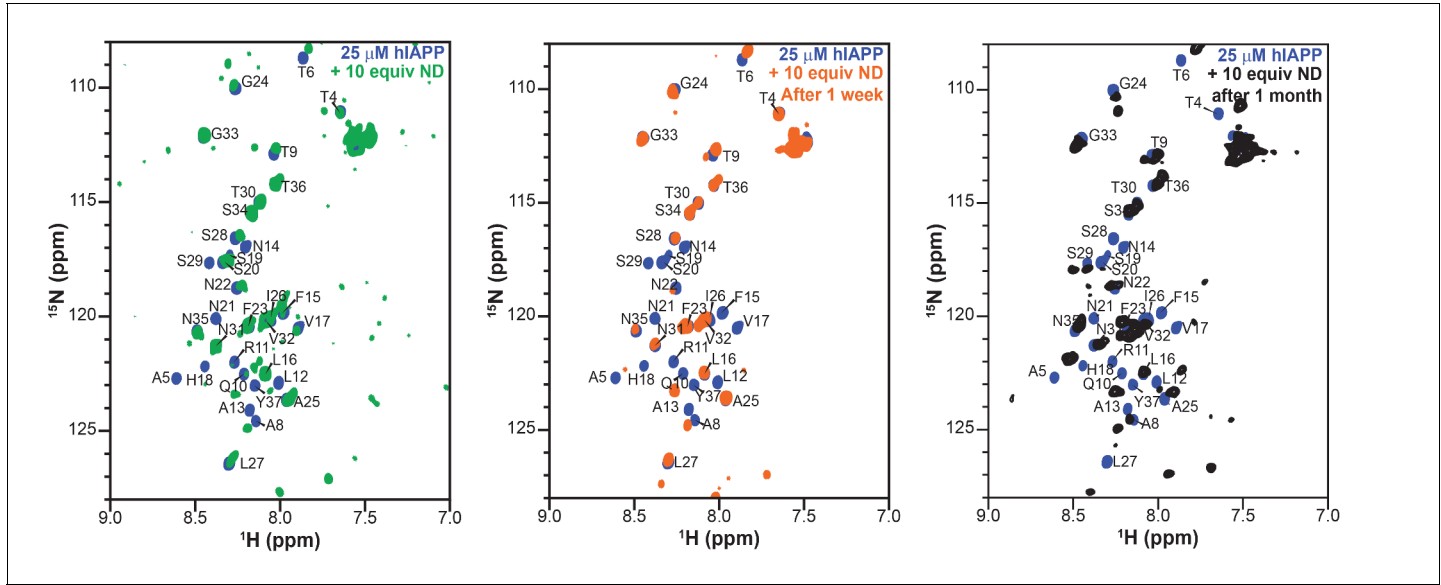

**Figure 4.** Stability and lifetime of NMR visible hIAPP-ND complexes. $^1H/^{15}N$ HMQC spectra were observable with only modest changes over the course of one month under quiescent conditions at room temperature, suggesting that the sample was amenable for very long (spanning to many days to a week) experiments that were needed for resonance assignment and structural characterization reported in this study.
DOI: https://doi.org/10.7554/eLife.31226.006

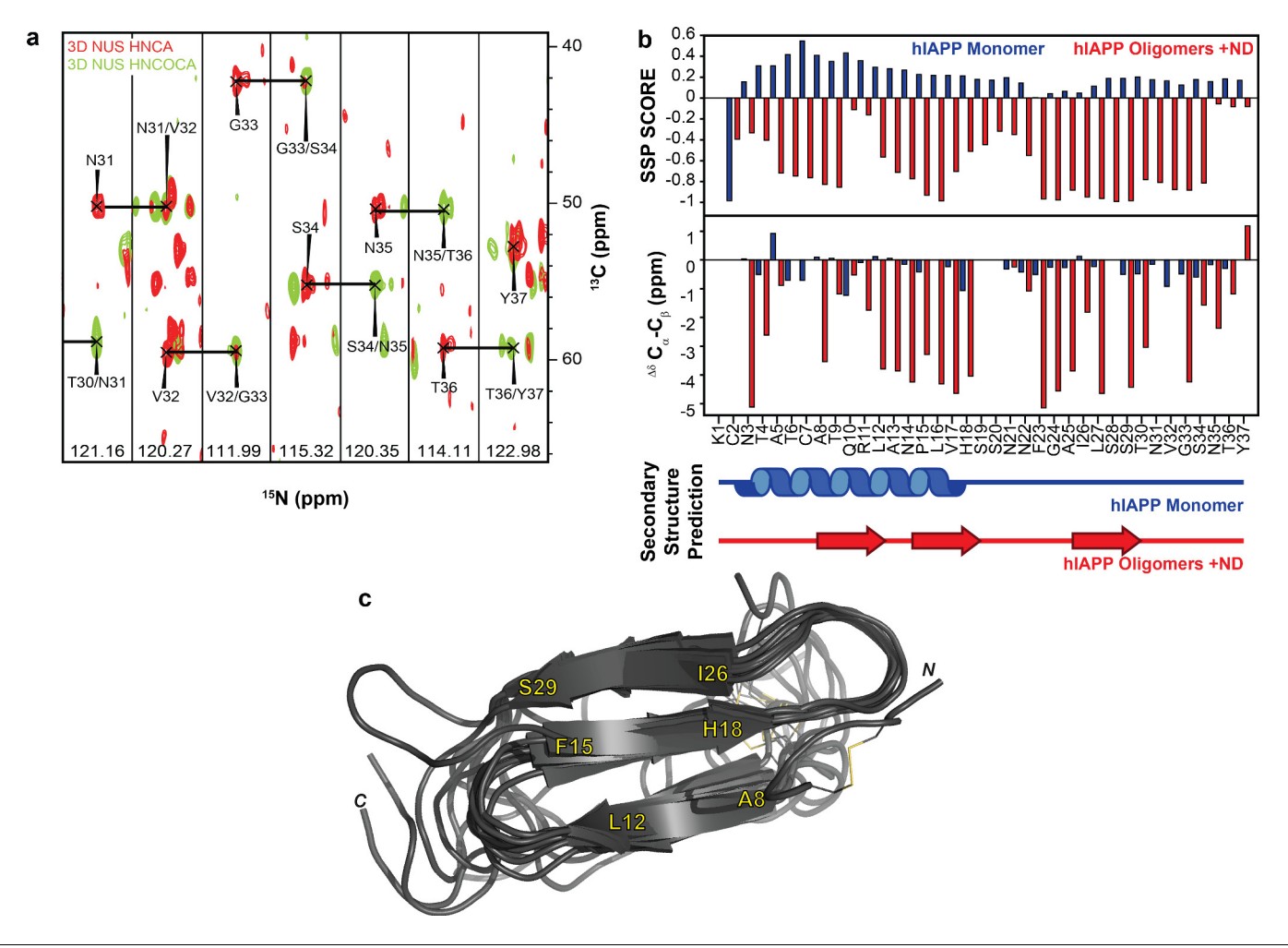

**Figure 5.** NMR characterization of hIAPP-ND1 interaction. (**a**) Triple-resonance (HNCA and HNCOCA) spectra of hIAPP oligomers in the presence of ND1 were utilized for resonance assignment and chemical shift determination (all strips can be found in *Figure 6*). (**b**) Secondary structure prediction performed using both Secondary Structure Propensity from Julie Forman-Kay's Laboratory and the $\Delta\delta$ $^{13}C\alpha$-$C_\beta$ secondary chemical shifts suggest a structure consisting of three β-strands (*Marsh et al., 2006*). (**c**) The 10 lowest energy structures were produced by CS-ROSETTA. The average $C_\alpha$-RMSD of lowest energy structure for residues 6–34 is 1.946 ± 0.521 and for all residues is 3.534 ± 0.489.

DOI: https://doi.org/10.7554/eLife.31226.007

region ($N^{22}FGAIL^{27}$) of hIAPP are located in the flexible loop regions of the model (*Westermark et al., 1990*). This suggests that these key residues may be accessible to other monomeric subunits in our model, indicating a possible mechanism of further aggregation for this folded intermediate when found outside of the constraints of the nanodisc. It is likely that this β-strand structure is influenced both by interaction with the lipid bilayer and interactions between monomeric subunits of a membrane-associated oligomer. However, inter-peptide contacts could not be observed in this approach, and therefore we are unable to estimate the size of the oligomer. Overall, these data represent the first non-fibrillar hIAPP structural model which contains β-strand secondary structure elements and the first ever experimentally derived, structural model of hIAPP interacting directly with an intact lipid bilayer.

## Oligomerization model and membrane orientation of hIAPP

With a structural model in hand, it is important to determine its membrane orientation to fully understand the roles of the intermediate structure as well as the lipid membrane. Unlike an amphipathic helical fold, a common feature of other amyloidogenic peptides interacting with a lipid bilayer, the

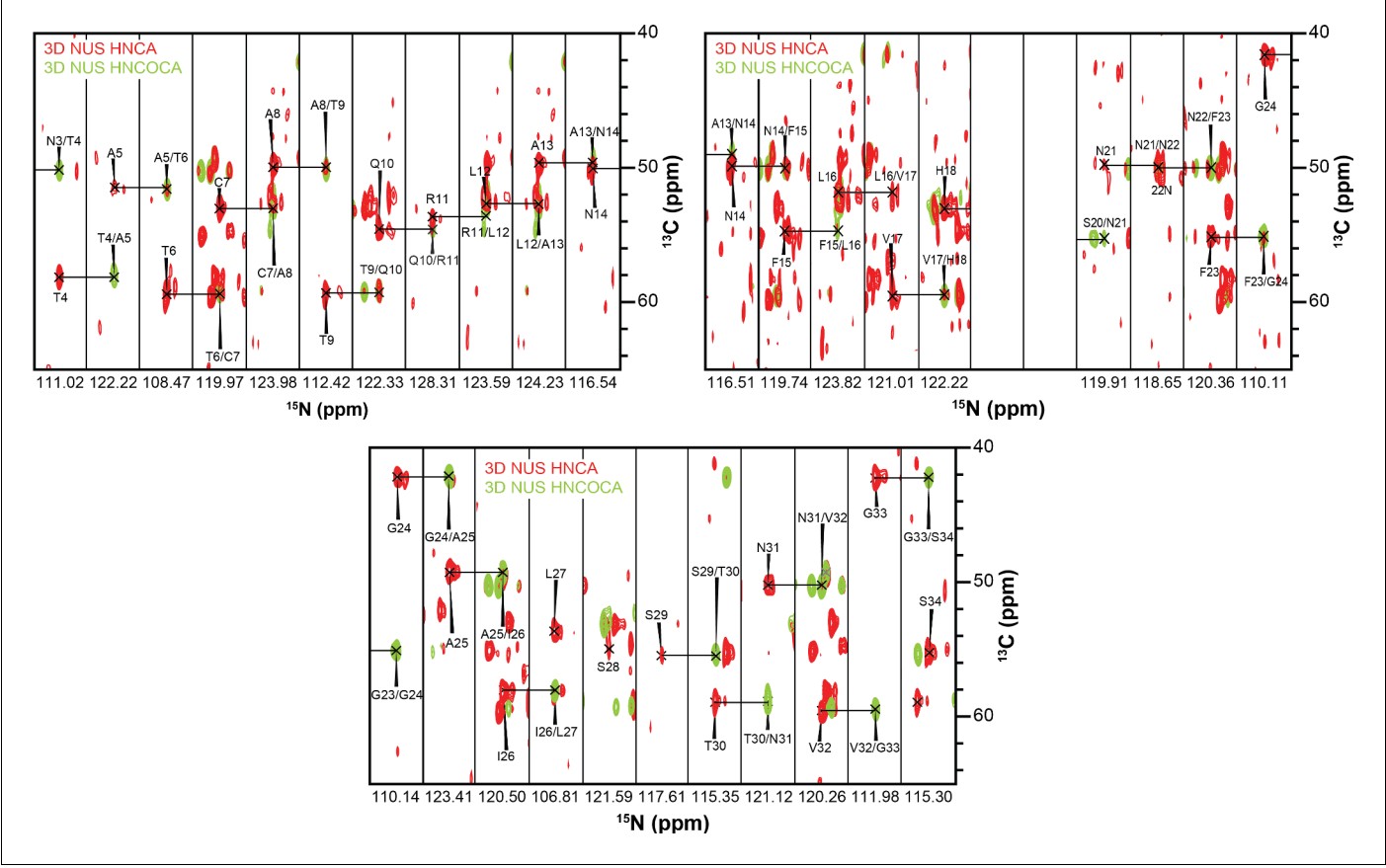

**Figure 6.** 3D strips used for resonance assignment. The majority of resonances from the 37 residues of hIAPP were resolvable in both HNCA and HNCOCA experiments that were performed on a 1:1 ratio of oligomeric-hIAPP:ND1. Chemical shift values were measured based on these resonance assignments for structural calculation reported in this study.

DOI: https://doi.org/10.7554/eLife.31226.008

proposed β-sheet structure of hIAPP does not possess explicit hydrophobic and hydrophilic surfaces (*Nath et al., 2011*). Therefore, we used NMR line broadening caused by binding to ND1 or paramagnetic probes to identify regions in the folded hIAPP species that directly interact with the membrane surface or the surrounding solvent (*Figure 8*).

The addition of ND1 to oligomeric hIAPP induced specific broadening of resonances due to direct interaction of residues with the lipid bilayer enhancing relaxation of resonances from the affected residues (*Figure 8a and d*). Broadening was predominantly observed for residues associated with the first two β-strands (R11, L12, V17, and H18) while residues in the unstructured N-terminus (T6) and the loop between the first two strands (A13). When mapped onto a surface model, these residues generally localize to a single region of the structure, suggesting a restricted site of interaction between the surface of ND1 and the folded structure. A Gd(III) solvent PRE complex was also titrated into a preformed complex of oligomeric hIAPP and ND1 in order to identify those residues most exposed (*Figure 8b and e*). Titration of a soluble $Gd^{3+}$ chelate affected an orthogonal set of resonances as compared to residues affected by binding to ND1 and are located in the loop between the second and third β-strands (F19, N22, F23, A25, and I26), as well as the disordered C-terminus (T30, N31, G33, S34, N35, T36 and Y37). These residues form two discrete surfaces at the ends of the modeled structure encompassing many of the unstructured residues located in the inter-strand loops, and they border the membrane-binding surface identified by titration with ND1, suggesting that these two distinct surfaces interact with high specificity with either the solution environment or the nanodisc. Finally, the membrane interaction region was confirmed by titration with 5-DOXYL steric acid (5-DSA) which preferentially quenches resonances located near the lipid bilayer

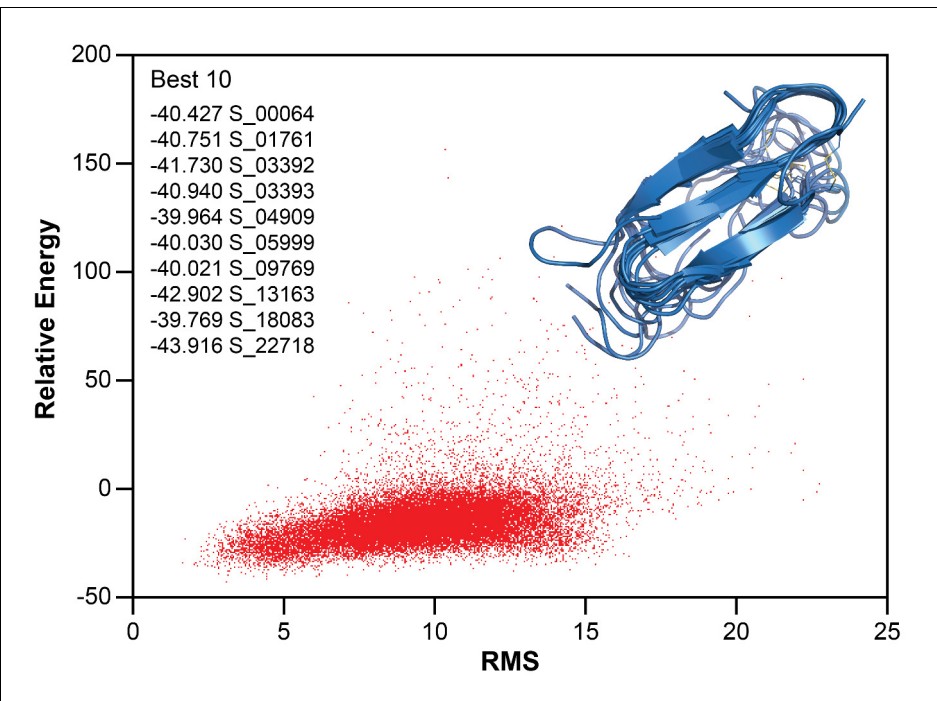

**Figure 7.** Structure calculation of membrane-associated hIAPP. The relative energy plot of the CS-Rosetta calculation, including an overlay of 10 lowest energy structures.

DOI: https://doi.org/10.7554/eLife.31226.009

surface (*Figure 8c and f*). Similar to the titration with ND1, 5-DSA selectively broadened residues in the first two β-strands, as well as the loop connecting the two strands, further confirming this region's preferential association with the lipid bilayer surface. Combined, these results suggest that the folded structure sits close to the bilayer surface with its β-sheet structure roughly perpendicular to the bilayer normal. The observed flexibility and solvent accessibility of the nucleating region of hIAPP (N$^{22}$FGAIL$^{27}$) suggest its availability to interact with other membrane-associated or soluble hIAPP species to promote the formation of higher ordered species (*Westermark et al., 1990*).

To better define the orientation of β-strand hIAPP intermediate within the lipid bilayer, molecular dynamics simulations with the Martini force field were performed for the intermediate in the presence of the lipid bilayer, and the findings were compared to a monomeric structure of hIAPP solved under identical solution conditions (*Figure 9*) (*Abraham et al., 2015*; *de Jong et al., 2013*; *Marrink et al., 2007*; *Rodriguez Camargo et al., 2017*). In both simulations, the monomeric subunit associated with the lipid bilayer. For the helical monomer, the helical N-terminus was strongly associated with the membrane while unstructured C-terminus was solvent exposed. For the β-strand intermediate, the N- and C-terminal residues are predominantly solvent accessible, along with residues N21 and N22 in the second loop region (*Nanga et al., 2011*). In both simulations, residues 11–19 have large interaction areas with the lipid but not the solvent, suggesting a possible site of initial interaction and structural conversion. The simulated results for the β-strand intermediate are in good agreement with NMR analysis of membrane and solvent interactions (*Figure 9e*). Both methods predict both the second loop region and the C-terminus to be flexible and solvent exposed while the residues in the first two β-strands are found to be membrane-associated. These findings further support the ability of the nucleating sequence, which resides in the second flexible loop, to promote inter-peptide interactions for the formation of larger, membrane-associated oligomers.

## Discussion

Although amyloid formation is common in many diseases and general principles underlying the folding pathways are understood, identifying and characterizing structural intermediates remains a major

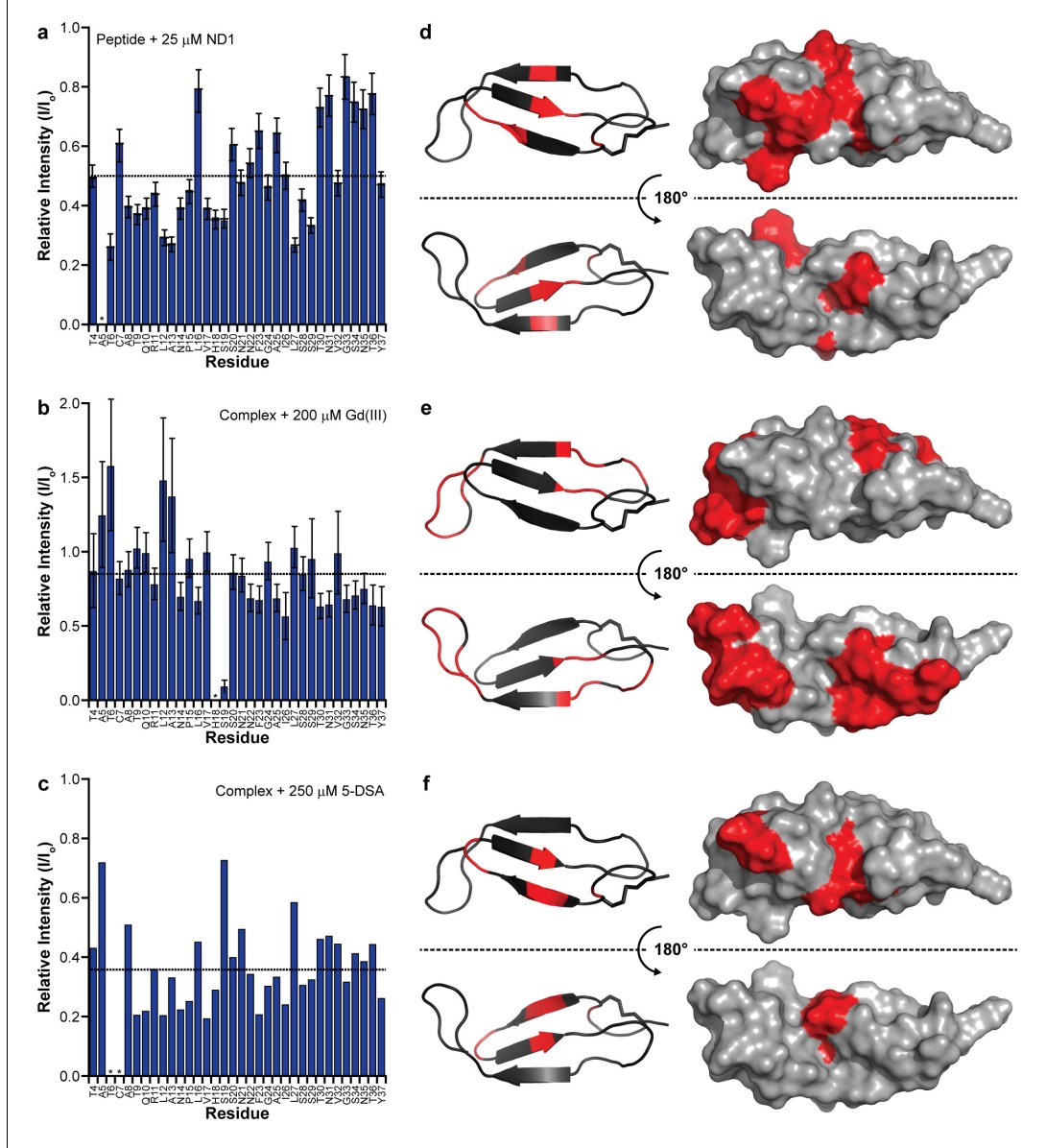

**Figure 8.** Identifying membrane-associated hIAPP interfaces. (a–c) Line broadening and signal reduction obtained from $^1$H-$^{15}$N HMQC spectra were used to identify the residues interacting directly with ND1 and compared to the average signal reduction for the sample (dashed line). (d–f) Highlighted in red are residues whose signal intensities were reduced more than the average and are mapped onto the structure. The addition of 1 equiv. ND1 (25 µM) to hIAPP identifies the residues directly interacting with the nanodisc surface (a,d) while the titration of Gd(III) (200 µM) into a solution of premixed hIAPP (50 µM) and ND1 (50 µM) selectively reduces the signal intensity of solvent accessible residues that are not interacting with ND1(b,e). Titrating 5-DSA (250 µM) into an identical sample containing a 1:1 ratio of hIAPP:ND1 selectively quenches the residues residing near the surface of ND1 (c,f).
DOI: https://doi.org/10.7554/eLife.31226.010

challenge. This difficulty is compounded when discussing amyloid formation in the presence of heterogeneous environments (or biomolecules) known to affect aggregation. Tools that can identify or stabilize unique intermediates are extremely valuable. Sequence- and conformation-specific antibodies have been developed as tools for basic research and potential therapeutics (*Kayed et al., 2010*; *Lee et al., 2016*; *Sevigny et al., 2016*). The development and discovery of small molecules capable of stabilizing and targeting distinct species of amyloid intermediates has been similarly pursued (*Doig and Derreumaux, 2015*; *Hamley, 2012*; *Pithadia et al., 2016*; *Young et al., 2015*).

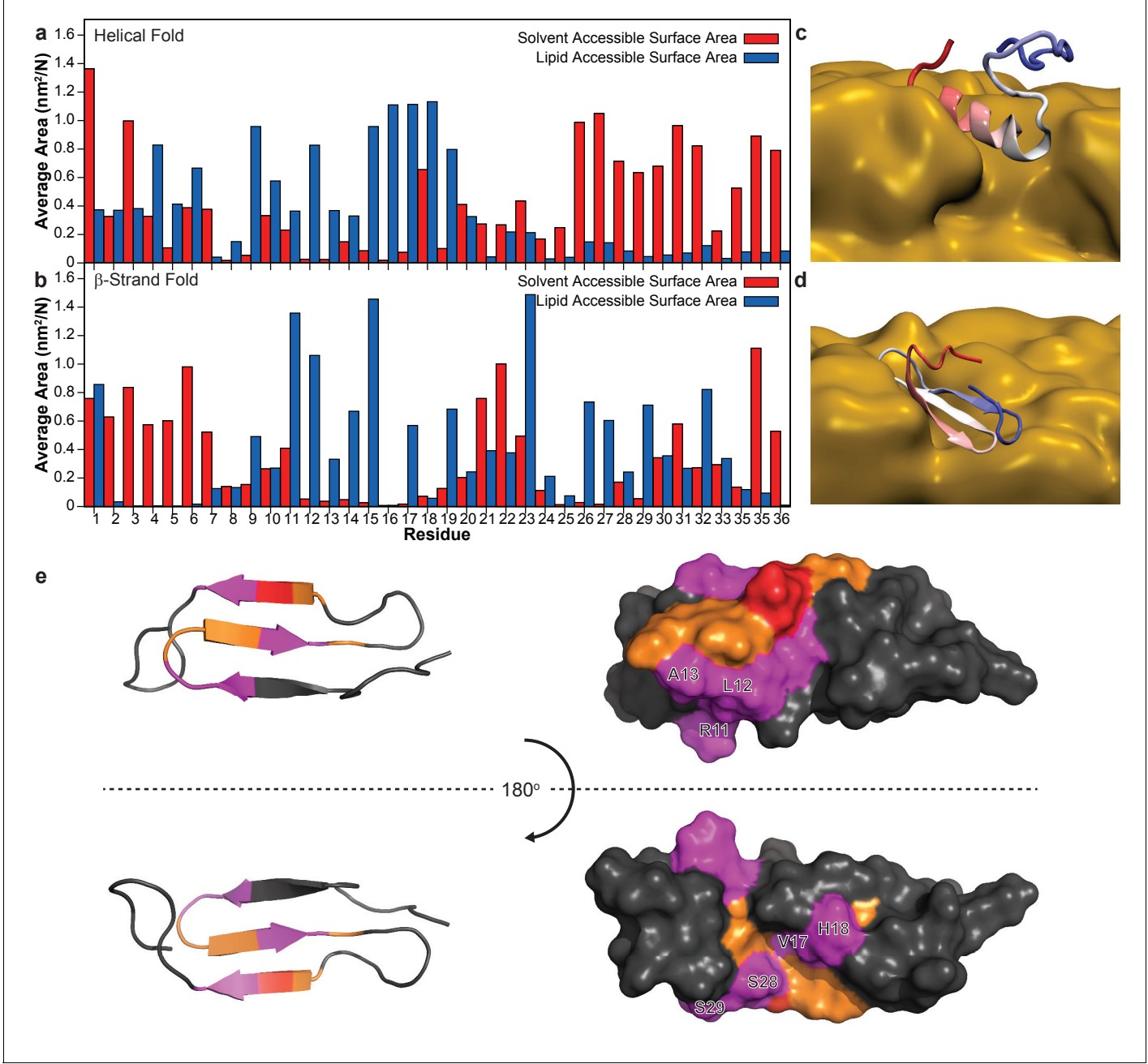

**Figure 9.** Simulation of membrane-hIAPP interactions. Molecular dynamic simulations with the Martini force field were performed to evaluate the preferential interactions of hIAPP with ND1 (*Abraham et al., 2015*; *de Jong et al., 2013*; *Marrink et al., 2007*). The average surface area of both a partially folded monomer (*Rodriguez Camargo et al., 2017*) (a) and the β-strand model (b) plotted for each residue. The preference for the solvent or lipid accessibility was mapped onto both simulated structures in the presence of the membrane (c,d). (e) When the hIAPP residues observed to interact with ND1 by NMR (red) are compared to those observed by MD simulation (orange), significant overlap is observed (magenta).
DOI: https://doi.org/10.7554/eLife.31226.011

Although these tools are capable of providing mechanistic insights into aggregation pathways, they continue to provide limited details regarding on oligomer structures.

Lipid nanodiscs represent a versatile tool to further the exploration of amyloid-membrane interactions with the potential to stabilize membrane-associated species within a confined space. Past work has utilized nanodiscs to investigate non-amyloidogenic sequences, amyloid-receptor interactions, and the impact of membrane composition on monomer affinity and has expanded our

understanding of the role of membranes and membrane proteins in amyloid-related biology (*Nath et al., 2011*; *Thomaier et al., 2016*; *Wilcox et al., 2015*). Herein, we have applied lipid nanodiscs to stabilize a membrane-associated intermediate of the amyloidogenic hIAPP for the first time. The isotropic nature of the nanodiscs facilitated the structural analysis of the stabilized species by conventional solution NMR which yielded a structural model of a non-fibrillar β-sheet intermediate. This folded model suggests a unique structure, unlike any previously reported results for hIAPP in either solution or the presence of membrane mimetics (*Figure 10*).

Previously, monomeric hIAPP in solution at pH 5.3, as well as in the presence of sodium dodecyl sulfate (SDS) micelles, was found to have a helical N-terminus, spanning residues T6-F15 (*Nanga et al., 2011*; *Rodriguez Camargo et al., 2017*). In the presence of SDS, hIAPP formed a second helical segment from S20-S29 while that same region is disordered in solution. Meanwhile, one fibrillar isoform of hIAPP formed in the absence of detergents or lipids contains a β-strand in the region where the monomeric form folded into an α-helix (*Luca et al., 2007*). The fiber's second β-strand encompassed I26-N35, overlapping partly with the second helical segment formed in SDS micelles. Interestingly, the two N-terminal β-strands of the hIAPP structure bound to ND (A8-L121 and F15-H18) overlap significantly with both the α-helical fold of the monomer and the first β-strand

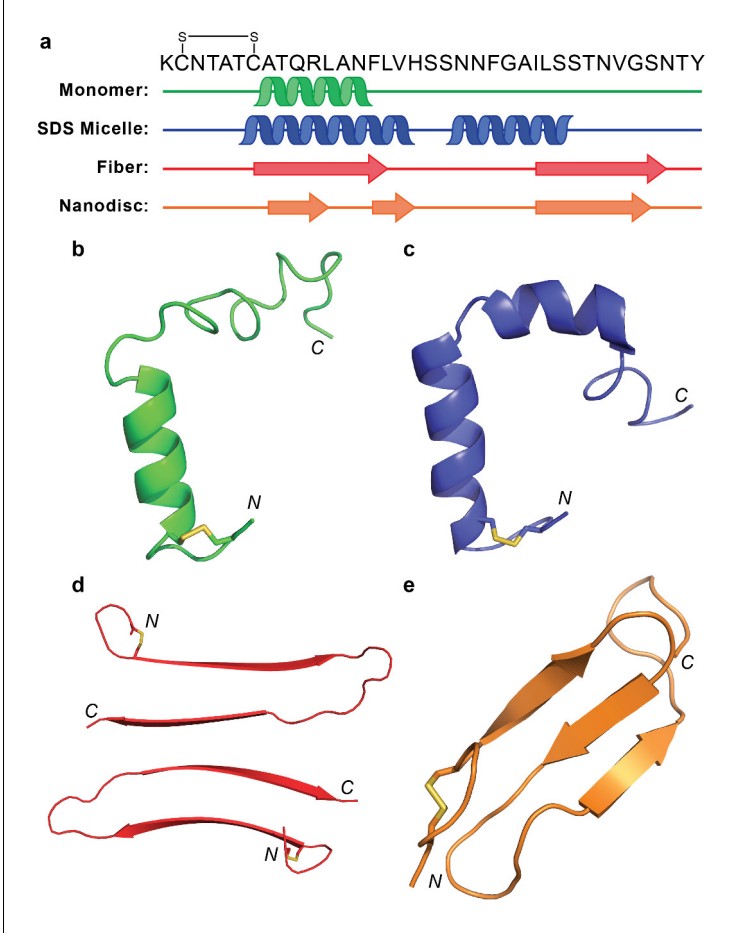

**Figure 10.** Comparison of hIAPP structures. (a) Known structures and models of hIAPP suggest partial folding of the monomeric subunit, although the folding varies with the sample preparation and environment. (b) Monomeric hIAPP prepared at pH 5.3 demonstrates a partial helical fold spanning C7-F15 (PDB: 5MGQ) (*Rodriguez Camargo et al., 2017*). (c) Monomeric hIAPP stabilized by SDS micelles adopts a similar N-terminal helix and a second helical region near the C-terminus (PDB: 2L86) (*Nanga et al., 2011*). (d) The striated ribbon morphology of hIAPP fibers shows twoβ-hairpins interacting through their C-terminal β-strands (*Luca et al., 2007*). (e) The folded hIAPP monomer interacting with the surface of ND1 possesses three antiparallel β-strands.
DOI: https://doi.org/10.7554/eLife.31226.012

of the fibrillar form. The ability of this sequence to adopt diverse secondary structures is surprising based on its predicted α-helical propensity from multiple sequence secondary structure prediction models (*Drozdetskiy et al., 2015*; *Raghava, 2002*). This highlights one of the fundamental difficulties in the study of amyloid structural intermediates: heterogeneous inter- and intramolecular interactions play a substantial role in promoting folding events. Both the fiber structure and membrane-associated model are capable of adopting the theoretically less favorable β-strand structure due to protein-protein and protein-lipid interactions, respectively. By comparing these four structural examples, however, it does appear that the N-terminal region consistently prefers to adopt some sort of secondary structure, rather than remain completely disordered. The extent of folding, however, is dependent upon external stimuli. It is interesting that the proposed nucleating sequence (N$^{22}$FGAIL$^{27}$) remains solvent exposed in all these models, supporting its role in promoting the interactions of distinct monomeric subunits of hIAPP in the process of amyloid formation.

The differences between these four structures highlight the challenges associated with structural characterization of hIAPP aggregation intermediates, as well as amyloid intermediates in general. Structure is highly dependent on the context, and those structures which can be observed need not be inherently relevant to the disease associated with the peptide of interest. This later point has been extensively explored in the evaluation of end stage amyloid fibril polymorphism (*LeVine and Walker, 2016*; *Stein and True, 2014*; *Tycko, 2015*). It is a problem likely to persist into the evaluation of oligomers. Nevertheless, it is imperative that the identification and interrogation of intermediate species continues. It is only through a greater breadth of structural information that correlations between structure and relevance can be made. To this end, as the first study to interrogate the structure of an hIAPP aggregation intermediate in the presence of a native lipid bilayer, we have demonstrated the value of nanodiscs in revealing structural details of membrane-associated aggregates. Through variation in lipid bilayer composition, nanodisc size, and aggregation conditions, it may be possible to stabilize and characterize a library of membrane-facilitated hIAPP aggregates. Through these studies, it is our hope that hallmarks of hIAPP oligomers may be identified. We suspect that subsequent studies of this system under different conditions (i.e. altered pH, membrane composition, and temperature) will, for instance, identify a maintenance of the flexible loop containing the self-recognition sequence. The lack of charged residues necessary to promote direct binding to the lipid membrane surface can enable flexibility and solvent exposure. This would reinforce the likelihood of this, or a similar structure, being relevant intermediates in the membrane-mediated aggregation of hIAPP. While this does not inherently translate to pathological relevance, it will provide further insight into the underlying mechanism of hIAPP's behavior, and possibly other amyloidogenic sequences as well. It is our hope that this overall approach will be translated to the study of other amyloidogenic peptides and proteins whose aggregation in a membrane environment may provide new insights into their toxicity and function. It must be noted that this methodology, while ideal for membrane-associated aggregation studies, has less value for the study of oligomers formed directly in solution as it remains unclear how the preformed oligomers may insert into a nanodisc. The mechanism of insertion into a nanodisc and the formation of oligomers in solution may less likely to be correlated. Therefore, this method yields limited insights to understand the general principles underlying protein aggregation. Our results may act as a blueprint to guide future structural investigations of membrane-associated amyloid species and shed light on the importance of these intermediates in amyloid-associated diseases. To accompany structural studies, experiments involving nanodiscs could be coupled with a variety of cutting edge NMR methodologies to investigate aspects of aggregation dynamics, intermediate size, and heterogeneity. Frosty (*Mainz et al., 2009*) or sedimentation NMR (*Bertini et al., 2013*) could be a useful tool to monitor the real-time size changes of nanodisc-associated oligomers, a method that would be intractable with conventional vesicle model membranes given their large size. Exchange-based methodologies such as CEST (*Fusco et al., 2016*) and DEST (*Fawzi et al., 2012*) could also be useful for the interrogation of lowly populated, rapidly or slowly exchanging folded or oligomeric intermediates. In addition, recently reported polymer-based nanodiscs and macro-nanodiscs, that uplifts the restriction on the size of lipid nanodiscs, could be used to apply a variety of solution and solid-state NMR experiments. Overall, nanodiscs represent a very powerful platform upon that can be employed to study intermediates formed in the process of protein aggregation.

## Materials and methods

### Recombinant hIAPP expression and purification

Full-length hIAPP (*KCNTATCATQRLANFLVHSSNNFGAILSSTNVGSNTY-NH₂, disulfide bridge 2–7*), both unlabeled and uniformly, isotopically labeled, was expressed following a previously described protocol (*Rodriguez Camargo et al., 2015*). Briefly, hIAPP is expressed in *E. coli* as a fusion with an N-terminal solubility tag and a C-terminal affinity tag. Following affinity purification, the C-terminal amide of native hIAPP is formed by incubating the fusion protein in a solution containing ammonium bicarbonate. The N-terminal solubility tag is then cleaved by V8 protease and the cleavage products are separated by filtration and reverse phase-HPLC. Finally, the disulfide bond is formed by treating the purified peptide with $H_2O_2$ in acetate buffer. Molecular biology reagents were obtained from New England Biolabs, Roche and from Sigma-Aldrich (St. Louis, MO). Isotopically labeled components for minimal media were purchased from Cambridge Isotope Laboratories (CIL).

### Nanodisc preparation

Nanodiscs have been assembled with a truncated version of MSP1D1, called MSP1D1ΔH5, as described previously (*Hagn et al., 2013*). A MSP-to-lipid ratio of 1:50 was used for DMPC (1,2-dimyristoyl-sn-glycero-3-phosphocholine) and DMPG (1,2-dimyristoyl-sn-glycero-3-phospho-(1'-rac-glycerol)) lipids. The percentage of negatively charged DMPG in the lipid blend was varied from 10% to 50%, as described in *Table 1*. All lipids were purchased from Avanti Polar Lipids (Alabaster, AL) or Cayman Chemical (Ann Arbor, MI). The final concentrations of MSP1D1ΔH5 was 200 µM, lipid concentration was 10 mM. Sodium cholate, that is required for lipid solubilization, was kept at a concentration of 20 mM in the assembly mixture in MSP-Buffer (20 mM Tris pH 7.5, 100 mM NaCl, 0.5 mM EDTA). After incubation for one hour at room temperature (RT), 0.7 g / mL of Biobeads-SM2 (Biorad) were added and the mixture was gently shaken for two more hours at RT. After removal of biobeads, the assembled nanodiscs were concentrated in an Amicon centrifugal device (50 kDa cut-off) (Merck-Millipore, Billerica, MA) to a final volume of 1 mL and purified on an S200a size excluz-sion column. One symmetric peak was obtained and concentrated to a 800 µL volume, yielding a nanodisc concentration of 260 µM (70% yield).

### Thioflavin-T assay

Amyloid aggregation kinetics in the presence of various nanodiscs were monitored by the amyloid-specific dye Thioflavin-T (ThT). Samples were prepared by initially dissolving unlabeled expressed hIAPP in a dilute HCl solution (pH 4) to a final concentration of 150 µM and maintained on ice. The peptide was further diluted into the appropriate buffer (either 20 mM $PO_4$, pH 7.4 or 30 mM Acetate, pH 5.3) in the presence of both 50 mM NaCl and 10 µM ThT to a final peptide concentration of 5 µM. The solutions also contained either 0, 0.5, 1, 2, 5, or 10 eq of either ND1, ND2, or ND3. Samples were subsequently plated in triplicate on uncoated Fisherbrand 96-well polystyrene plates and readings were taken on a Biotek Synergy two microplate reader. Samples were incubated for 48 hr in the instrument at either 25 or 35°C with continuous, slow orbital shaking. Wells were read from the bottom with an excitation wavelength of 440 nm (30 nm bandwidth) and an emission wavelength of 485 nm (20 nm bandwidth) at 4-min intervals.

Following data acquisition, the raw fluorescence traces were background corrected and normalized. Normalized curves were subsequently fit to Eq. q and *Equation 2* in order to calculate the lag time ($t_{lag}$) for each curve (*Batzli and Love, 2015*). The $t_{lag}$ values for each experimental condition were subsequently averaged across three separate trials.

$$F(t) = F_{inf} + \frac{F_0 - F_{inf}}{(1 + e^{k(t - t_{50})})} \qquad (1)$$

$$t_{lag} = t_{50} - \frac{2}{k} \qquad (2)$$

### Transmission electron microscopy (TEM)

Samples of freshly purified ND1 (50 µM), ND1incubated with hIAPP (50 µM of each) for one week, and fibers formed by incubating 50 µM hIAPP in buffer for one week were prepared. TEM grids

were prepared by adding 10 µL of the sample were placed on the grid (Formvar/Carbon 300 mesh copper coated grids from Electron Microscopy Sciences) for 1 min followed by the removal of excess liquid by filter paper. Grids were then stained for 2 min with 10 µL of 1% uranyl acetate solution. The excess liquid was again dried using filter paper. The grid was again treated with 10 mL of 1% uranyl acetate solution for 30 s before the liquid was dried with filter paper. Samples were immediately measured on Transmission Electron Microscopy employing a Zeiss EM 10 CR (Zeiss, Germany).

## Size exclusion chromatography (SEC)

SEC was performed on samples of freshly prepared nanodisc, monomeric hIAPP, oligomeric hIAPP, and mixtures of hIAPP and nanodiscs with a flow rate of 0.5 mL/min on an Äkta Pure protein purifier (GE Healthcare) using a semi-preparative Superdex S200 Increase 10/300 GL column (24 mL bed volume, GE Healthcare) equilibrated in MSP buffer.

## Gel electrophoresis

Tricine-SDS-PAGE gel electrophoresis was performed using 16% Tris-tricine-SDS gels (*Schägger, 2006*). Gels was run at 10W for approximately 1 hr, followed by fixation for 10 min in a solution of 50% methanol and 20% acetic acid. The gels were stained for 10 min with a 0.25% solution of the dye Coomassie (Serva) dissolved in 15% methanol and 10% acetic acid. Gels were subsequently washed with de-ionized water, and destained for 10 min with 10% acetic acid. The final gel was stored in water and imaged.

## NMR sample preparation

A lyophilized aliquot of expressed hIAPP was dissolved into the NMR buffer containing 30 mM deuterated Acetate (pH 5.3) with 10% $D_2O$. After NMR measurements, the samples were measured and stored at 4°C when the peptide was along. However, to work with the nanodiscs, we stored the samples at room temperature and NMR experiments were carried out at 35°C. The formation of the intramolecular disulfide bond was confirmed by NMR. The formation of the intramolecular disulfide bond was confirmed by NMR. To perform NMR experiments with the nanodiscs. The hIAPP powder was dissolved in a small amount of buffer, vortexed and mixed with a concentrated solution of ND to the desired final ratio, completed the final volume with buffer until 250 µM. The final hIAPP concentration was 50 µM in all cases. Samples were directly transferred into a Shigemi NMR tubes (Shigemi Inc., Allison Park) for NMR measurements. In experiments utilizing the monomeric preparation of hIAPP, peptide expression, purification, and oxidation were completed one day prior to starting NMR measurements. The freshly prepared hIAPP peptide was dissolved directly into buffer right before beginning NMR data acquisition. In contrast, hIAPP oligomers were generated from expressed, purified, and oxidized peptide prepared 4 weeks prior to experimental measurement, but the peptide was stored at −20°C which allowed the semi-hydrated peptide powder to form an early aggregate. ESI-MS was performed to ensure that no degradation occurred during oligomer formation.

## NMR experiments

NMR experiments employing Bruker Avance 500, 600, 750 MHz spectrometers were performed at 35°C. The proton chemical shifts were referenced to the water resonance frequency while the $^{15}N$ and $^{13}C$ shifts were referenced indirectly. Backbone and side chain assignments were obtained using triple resonance experiments HNCA and HNCOCA. (*Sattler et al., 1999*) Side-chain assignments and chemical shifts were obtained from $^{13}C$ HSQC assignment transposition. Overall, an assignment completeness of 97% was obtained. NMR spectra were processed using the software TopSpin (Bruker) and NMRPIPE (*Goddard and Kneller, 1997*). Spectra were analyzed using ccpNMR analysis (*Vranken et al., 2005*).

## Structural model calculation

A structural model for the folded subunit of hIAPP was calculated using the Chemical-Shift-ROSETTA (CS-ROSETTA Version 4.8) server from the Biological Magnetic Resonance Data Bank Rosetta. CS-ROSETTA is a robust tool for de novo protein structure generation, using $^{13}C$, $^{15}N$ and $^1H$ NMR chemical shifts as input. The CS-ROSETTA approach utilizes SPARTA-based selection of protein

fragments from the PDB, in conjunction with a regular ROSETTA Monte Carlo assembly and relaxation procedure, to generate structures of minimized energies. In addition, an alternative CS-ROSETTA fragment selection protocol is provided that improves robustness of the method for proteins with missing or erroneous NMR chemical shift input data (*Lange et al., 2012*; *Shen et al., 2008*, *Shen et al., 2010*, *Shen et al., 2009*).

### Molecular dynamics simulation

Molecular dynamics (MD) simulations using Gromacs 5.1.2 were performed to determine the membrane interaction and orientation of the peptide (*Abraham et al., 2015*). Two peptide models based on NMR determined structures were created: a helix fold and the β-strand intermediate structure of hIAPP. To describe the protein interaction, the Martini force field version 2.2 was used together with an elastic network to conserve the secondary structure information (*de Jong et al., 2013*; *Marrink et al., 2007*). To mimic the experimental condition, a pH value of 5.3 was taken into account by neutralizing the N and C termini and placing a positive charge on the His-18 side chain. A 9:1 DMPC:DMPG lipid bilayer was created using the insane script (*Wassenaar et al., 2015*) and Martini 2.0 lipids parameters. The standard Martini water model was used (*Marrink et al., 2007*).

Both systems were run in the isothermal-isobaric (NpT) ensemble using 30 fs time steps, a temperature of 300 K, and a pressure of 1 bar. In both cases, the peptide was initially positioned in solution. The simulation length was 25 µs for the helix fold and 10 µs for the β-strand structure. To control the temperature, the v-rescale thermostat was used with a coupling constant $\tau_t = 1$ (*Bussi et al., 2007*). The pressure was semi-isotropic coupled with a coupling constant of $\tau_p = 20$ ps and a compressibility of $\chi = 3.0 \times 10^{-4}$ $bar^{-1}$ using the Parrinello-Rahman barostat (*Parrinello and Rahman, 1981*). The Verlet cutoff-scheme was used for the calculation of the electrostatic and the van der Waals interactions with a cut-off of 1.1 nm and dielectric constant of 15. The same starting box size of 10 nm x 10 nm x 15 nm and the same amount of membrane molecules (304 DLPC, 32 DLPG) were used in both setups.

The Gromacs SASA tool (*Abraham et al., 2015*; *Eisenhaber et al., 1995*) was used to calculate the average solvent accessible surface area (SASA) per residue of the peptide bonded to the membrane surface. A higher van der Waals distance of 0.21 nm was used to account for the Martini force field. The lipid accessible surface area was calculated as difference of the peptide SASA and the SASA of the peptide-membrane system.

## Acknowledgements

This work was supported by funds from NIH (to AR), a Hans-Fischer Senior Fellowship of the Institute of Advanced Studies of the Technische Universität München awarded to AR, and partly by the Protein Folding Initiative at The University of Michigan (to AR). We acknowledge support from the Helmholtz-Gemeinschaft and the Deutsche Forschungsgemeinschaft (DFG, grants Re1435 and SFB-1035, project B07 (to BR) and B13 (to FH). We are further grateful to the Center for Integrated Protein Science Munich (CIPS-M) for financial support (to BR and FH). AJ, FH and CC acknowledge the support of the Technische Universität München – Institute for Advanced Study, funded by the German Excellence Initiative and the European Union Seventh Framework Program under grant agreement no. 291763. The authors gratefully acknowledge the Gauss Centre for Supercomputing e.V. (www.gauss-centre.eu) for funding this project by providing computing time on the GCS Supercomputer SuperMUC at Leibniz Supercomputing Centre (www.lrz.de).

## Additional information

### Funding

| Funder | Grant reference number | Author |
| --- | --- | --- |
| National Institutes of Health | AG048934 | Ayyalusamy Ramamoorthy |

The funders had no role in study design, data collection and interpretation, or the decision to submit the work for publication.

## Author contributions

Diana C Rodriguez Camargo, Formal analysis, Validation, Investigation, Methodology, Writing—original draft; Kyle J Korshavn, Formal analysis, Investigation, Visualization, Methodology, Writing—original draft; Alexander Jussupow, Kolio Raltchev, David Goricanec, Markus Fleisch, Riddhiman Sarkar, Kai Xue, Michaela Aichler, Gabriele Mettenleiter, Investigation; Axel Karl Walch, Resources, Investigation; Carlo Camilloni, Resources, Software, Formal analysis, Investigation, Writing—review and editing; Franz Hagn, Conceptualization, Resources, Formal analysis, Investigation, Methodology, Writing—review and editing; Bernd Reif, Conceptualization, Resources, Supervision, Funding acquisition, Investigation, Methodology, Writing—review and editing; Ayyalusamy Ramamoorthy, Conceptualization, Resources, Supervision, Funding acquisition, Investigation, Methodology, Project administration, Writing—review and editing

## Author ORCIDs

Diana C Rodriguez Camargo (iD) https://orcid.org/0000-0003-3522-4859
Carlo Camilloni (iD) http://orcid.org/0000-0002-9923-8590
Ayyalusamy Ramamoorthy (iD) http://orcid.org/0000-0003-1964-1900

## Decision letter and Author response

Decision letter https://doi.org/10.7554/eLife.31226.015
Author response https://doi.org/10.7554/eLife.31226.016

## Additional files

### Supplementary files

• Transparent reporting form
DOI: https://doi.org/10.7554/eLife.31226.013

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
