## [Decision Letter]

Thank you for submitting your article "Stabilization and structural analysis of a membrane-associated hIAPP aggregation intermediate" for consideration by *eLife*. Your work has been evaluated very favorably by three experts, one of whom is a member of our Board of Reviewing Editors, who felt that it is an interesting and highly relevant paper. The evaluation has been overseen by a Reviewing Editor and John Kuriyan as the Senior Editor.

The reviewers have discussed the reviews with one another and the Reviewing Editor has drafted this decision to help you prepare a revised submission

The significance of this manuscript is two-fold. First, it provides new insight into how IAPP interacts with membranes, showing that in the membrane-associated intermediate investigated here the nucleating NFGAIL sequence remains flexible and solvent exposed and hence available for interactions with IAPP monomers. Second, this manuscript proposes a widely applicable experimental strategy to overcome the challenges typically encountered when attempting to extract structural information on amyloid intermediates, which are typically transient and polymorphic. Hence, this paper will be of interest to a wide range of readers, particularly in the field of structural biology, biophysics, neuroscience and medicinal chemistry. We do ask that you respond to the issues raised below in a revised version. Please use your own judgement in deciding how best to respond to each point.

1) The authors speak somewhat briefly about the fact that it is important to obtain many structures under different conditions and try to establish general principles, given that structures may change somewhat depending on environment. In this sense, therefore, how 'robust' do the authors feel that their structure really is – that is how sensitive is it to conditions. Although it is beyond the scope to expect additional structures to be determined under diverse conditions, the authors might be able to comment how pH plays a role or nanodisc composition. Just a qualitative feel would be good here. In this context, is there any evidence that the trapped intermediates are physiological?

2) There appears to be little binding of the monomer, yet binding of a small oligomer. Since the evidence is that only a single structure is associated with the disc can the authors provide some insight into why monomers would not be able to bind. Do the oligomers dissociate into monomers upon binding? Can the authors be certain that they are focusing on signals only from the bound state and not any unbound monomer?

3) In the PRE study some of the relative intensities are greater than 1 by a considerable amount. Can the authors speculate about this? There are no error bars in this figure and others like it in Figure 8 and so I wonder about the quality of the data and hence interpretations. Please include error bars.

4) Do the molecular dynamics results agree with the results of binding in terms of the positioning of the peptide on the disc? Perhaps a comparison of color coded structures showing regions of interactions with the disc as established by MD and by the NMR study of Figure 8 would be illustrative.

5) Is it possible that Tht interacts with nanodiscs and if so, would such interaction affect the Tht fluorescence and/or the amount of Tht available to interact with cross-β IAPP structures? Addressing these questions through control experiments and/or data analysis would enhance the interpretation of Figure 1 and Figure 2.

6) The authors report that no appreciable chemical shift differences are observed between the 1H-15N HMQC spectra of the monomers and oligomers. This is likely because only monomers are observable in the oligomer sample. However, addition of ND to the monomeric sample causes only minimal chemical shift changes, whereas addition of ND to the oligomeric sample results in significant chemical shift variations. Can the authors rationalize why the NMR detectable species of the monomeric and oligomeric samples appear similar in the absence of nanodiscs, but significantly different in the presence of nanodiscs? Why are the monomers unable to form the trapped oligomeric intermediate in the presence of nanodiscs?

7) Often trapped oligomeric intermediates are kinetically but not thermodynamically stable. In this case, the order of addition of the reagents in the mixture is critical. Were the nanodiscs added before or after the ~4 weeks incubation period required to generate oligomers? This should be explained clearly in the Material and methods section.

8) We carefully compared Figure 3 vs. Figure 5 and these two figures appear identical to us. Why is the same Figure shown twice? Why is Figure 3 referred to as a 2D compression of an HNCA, while Figure 5 is referred to as an HSQC? We would expect the two spectra to be perhaps similar, but not identical (especially at the level of noise).

9) The authors should explain why in several cases they decided to acquire HN-HMQC vs. HN-HSQC spectra. The effective MW of the IAPP oligomer – ND complexes is presumably quite high. Why not HN- or even better Methyl-TROSY?

10) The authors should clarify the merits of complementing 2D experiments (e.g. HMQC) with 2D projections of 3D experiments (e.g. HNCA)

11) While the proposed intermediate trapping methods based on nanodiscs is ingenious and elegant, the discussion should emphasize that it is currently unknown to what extent the nanodisc trapped intermediate reflects the oligomers populated in solution in the absence of lipids.

---

## [Author Response]

The significance of this manuscript is two-fold. First, it provides new insight into how IAPP interacts with membranes, showing that in the membrane-associated intermediate investigated here the nucleating NFGAIL sequence remains flexible and solvent exposed and hence available for interactions with IAPP monomers. Second, this manuscript proposes a widely applicable experimental strategy to overcome the challenges typically encountered when attempting to extract structural information on amyloid intermediates, which are typically transient and polymorphic. Hence, this paper will be of interest to a wide range of readers, particularly in the field of structural biology, biophysics, neuroscience and medicinal chemistry. We do ask that you respond to the issues raised below in a revised version. Please use your own judgement in deciding how best to respond to each point.1) The authors speak somewhat briefly about the fact that it is important to obtain many structures under different conditions and try to establish general principles, given that structures may change somewhat depending on environment. In this sense, therefore, how 'robust' do the authors feel that their structure really is – that is how sensitive is it to conditions. Although it is beyond the scope to expect additional structures to be determined under diverse conditions, the authors might be able to comment how pH plays a role or nanodisc composition. Just a qualitative feel would be good here. In this context, is there any evidence that the trapped intermediates are physiological?

As the reviewers note, we have been cautious to define the structure of human-IAPP that we have elucidated in this study as a definitive, disease relevant intermediate. Given the considerable ambiguity within the amyloid field as to the relevance of oligomers, stability of toxic intermediates, and mechanism of action, we do not wish to over interpret the functional impact of our structural study. To reinforce this view, we have expanded our discussion at the end of the paper as follows: “We suspect that subsequent studies of this system under different conditions (i.e. altered pH, membrane composition, and temperature) will, for instance, identify the maintenance of the flexible loop containing the self-recognition sequence. The lack of charged residues necessary to promote direct binding to the lipid membrane surface can enable flexibility and solvent exposure. This would reinforce the likelihood of this, or a similar structure, being relevant intermediate in the membrane-mediated aggregation of hIAPP. While this does not inherently translate to pathological relevance, it will provide further insight into the underlying mechanism of hIAPP’s behavior, and possibly other amyloidogenic sequences as well.

2) There appears to be little binding of the monomer, yet binding of a small oligomer. Since the evidence is that only a single structure is associated with the disc can the authors provide some insight into why monomers would not be able to bind. Do the oligomers dissociate into monomers upon binding? Can the authors be certain that they are focusing on signals only from the bound state and not any unbound monomer?

We apologize for the initial ambiguity in our analysis pertaining to this point. As noted in the original version of the manuscript, the minimal chemical shift change upon the addition of ND1 to monomeric hIAPP “suggests that only a small portion of the NMR visible hIAPP population in the monomeric preparation stably interacts with ND1 during the time scale of the NMR experiment (~ 1 hour)”. We have updated this section in the revised manuscript to read as follows: “This suggests that only a small portion of the NMR visible hIAPP population in the monomeric preparation stably interacts with ND1 within the duration of the NMR experiment (~ 1 hour); monomeric hIAPP undoubtedly binds to ND1, however the exchange rate of the highly dynamic process is too rapid to result in detectible spectral changes.”

Additionally, as noted in the discussion, the structure reported in this study was collected under identical solution conditions (pH 5.3, 30 mM acetate) as used in a recent study that reported a monomeric structure of hIAPP in the absence of lipids. A comparison of the chemical shifts observed for the reported solution hIAPP structure with that presented in this study gives us assurance that here we are observing a unique species and not simply unbound monomer which would adopt identical chemical shifts to the previously reported monomeric structure. We believe that the oligomer is the observed species in this study as it is not in rapid exchange with ND1, unlike the monomer, and thus the species is sufficiently stable and long-lived to provide homogeneous NMR signals.

3) In the PRE study some of the relative intensities are greater than 1 by a considerable amount. Can the authors speculate about this? There are no error bars in this figure and others like it in Figure 8 and so I wonder about the quality of the data and hence interpretations. Please include error bars.

We apologize for this oversight and thank the reviewers for this comment. Error bars are added to Figure 8 in the revised manuscript. While solvent PRE is a useful tool to study protein solvent interactions, using amide chemical shift to monitor the solvent accessibility present a complication due to the rapidly exchanging hydroxyl protons of residues such as Ser, Thr and Tyr. This rapid exchange can introduce significantly more deviation in the measurement of the exchangeable residues and those which surround it. As a result, we have not used PRE values to make quantitative structural analysis. Instead we have used the PRE results as general indicators of trends for bound or unbound.

4) Do the molecular dynamics results agree with the results of binding in terms of the positioning of the peptide on the disc? Perhaps a comparison of color coded structures showing regions of interactions with the disc as established by MD and by the NMR study of Figure 8 would be illustrative.

We thank the reviewers for this observation. To answer this question and show the agreement of the results, we updated Figure 9 to include a panel highlighting the interaction region obtained by MD simulation in orange and the results obtained by NMR in red, with the shared residues highlighted in magenta. The interacting regions are very similar, however, not identical. Discrepancies between the NMR and simulations are described in the main text of the manuscript.

5) Is it possible that Tht interacts with nanodiscs and if so, would such interaction affect the Tht fluorescence and/or the amount of Tht available to interact with cross-β IAPP structures? Addressing these questions through control experiments and/or data analysis would enhance the interpretation of Figure 1 and Figure 2.

We thank the reviewers for this observation and have updated Figure 1 to share some of our control results. Figure 1 now demonstrates, that ThT fluorescence in buffer is comparable to fluorescence in the presence of ND in solution. Additionally, conventional fibrillar aggregates formed in solution achieve similar maximal fluorescence values in comparison to those formed in the presence of ND, which confirms that the fibrillation is not inhibited by interactions with the ND surface. This is explained in detail in the revised manuscript. Use of nanodiscs results in only a minimal reduction of S/N ratio in the experiment, which is a great advantage in the study of amyloid-membrane interactions.

6) The authors report that no appreciable chemical shift differences are observed between the 1H-15N HMQC spectra of the monomers and oligomers. This is likely because only monomers are observable in the oligomer sample. However, addition of ND to the monomeric sample causes only minimal chemical shift changes, whereas addition of ND to the oligomeric sample results in significant chemical shift variations. Can the authors rationalize why the NMR detectable species of the monomeric and oligomeric samples appear similar in the absence of nanodiscs, but significantly different in the presence of nanodiscs? Why are the monomers unable to form the trapped oligomeric intermediate in the presence of nanodiscs?

Like the reviewers we were surprised by the differences in the spectra of the monomeric and oligomeric preparations of hIAPP in the absence and presence of ND1. It is known that freshly dissolved hIAPP has a small subpopulation of oligomeric species, though their structure and distribution are not well understood. We believe that our oligomeric preparation enriches this population, though the monomeric form of hIAPP remains predominant. When the two different preparations were introduced to ND1, monomeric hIAPP has weak, transient interactions with ND1, as detailed earlier, while the oligomer which is enriched by pre-incubation in solution binds more tightly and shifts the equilibrium towards the oligomeric population in solution, which facilitates further binding. In essence, ND1 acts as a sink for the enriched oligomeric population to drive additional oligomer formation. As this population becomes enriched, it dominates the NMR spectrum. We have not yet attained experimental insight into the driving force which promotes this specific oligomer to interact with the membrane preferentially and is an ongoing area of study that merits its own publication.

7) Often trapped oligomeric intermediates are kinetically but not thermodynamically stable. In this case, the order of addition of the reagents in the mixture is critical. Were the nanodiscs added before or after the ~4 weeks incubation period required to generate oligomers? This should be explained clearly in the Material and methods section.

We agree completely with the reviewers that the order upon which reagents are added is critical for the obtained results. We have explained this in the revised manuscript in detail. To produce oligomers, we incubated hIAPP for approximately four weeks without nanodiscs. The nanodiscs were subsequently added, directly prior to the NMR experiment (excluding the NMR titration experiment). The Materials and methods section has been updated to clarify this point.

8) We carefully compared Figure 3 vs. Figure 5 and these two figures appear identical to us. Why is the same Figure shown twice? Why is Figure 3 referred to as a 2D compression of an HNCA, while Figure 5 is referred to as an HSQC? We would expect the two spectra to be perhaps similar, but not identical (especially at the level of noise).

We apologize of this mistake. We have corrected the error by removing the duplicated spectrum from Figure 5.

9) The authors should explain why in several cases they decided to acquire HN-HMQC vs. HN-HSQC spectra. The effective MW of the IAPP oligomer – ND complexes is presumably quite high. Why not HN- or even better Methyl-TROSY?

A principle limitation in our study was sample concentration. While we began with a sample of 50 μM, only 70-80% was stable over the course of an experiment, leaving ~40 μM peptide to be observable. We therefore elected to use HMQC given its enhanced sensitivity over HSQC (up to three-fold increase). The sensitivity gains can be traced to the differential relaxation of the transitions that gives rise to the way in which magnetization is transferred in each of the experiments and the correlations. In practice, TROSY has the potential to reduce the fast decay of magnetization due to T2 relaxation (i.e. spin-spin relaxation) by constructive use of the two effects that mainly determine transverse relaxation in large proteins at high magnetic field strengths. These are dipole-dipole coupling between two spins, and chemical shift anisotropy of the two associated nuclei. The TROSY method selects the slowest relaxing component of the multiplet, resulting in a loss of intensity for fast tumbling molecules. We found that TROSY type experiments to be not beneficial for the systems investigated in this study. Empirically, we obtained the best results using a standard HNCA in combination with Non Uniform Sampling (NUS). We did not pursue methyl-TROSY experiments, as we were able to record HMQC and 3D HNCA experiments by overcoming the challenges by the difficult sample.

10) The authors should clarify the merits of complementing 2D experiments (e.g. HMQC) with 2D projections of 3D experiments (e.g. HNCA)

Our goal in comparing 2D NMR experiments with 2D projections of 3D experiments was to qualitatively evaluate the status of the peptide’s resonances throughout the course of the assignment experiment, while using the 2D experiments as a reference. In situations when we observed that many of the peaks had disappeared in the 2D projection, we decided no longer to pursue structural characterization of what was likely a highly heterogeneous or unstable sample.

11) While the proposed intermediate trapping methods based on nanodiscs is ingenious and elegant, the discussion should emphasize that it is currently unknown to what extent the nanodisc trapped intermediate reflects the oligomers populated in solution in the absence of lipids.

We thank the reviewers for this comment and the observation is included in the Discussion section. It reads as given below: “It must be noted that this methodology, while ideal for membrane-associated aggregation studies, has less value for the study of oligomers formed directly in solution as it remains unclear how the preformed oligomers may insert into a nanodisc. The mechanism of insertion into a nanodisc and the formation of oligomers in solution may less likely to be correlated. Therefore, this method yields limited insights to understand the general principles underlying protein aggregation.”